# Mobile Health Interventions to Improve Health Behaviors and Healthcare Services among Vietnamese Individuals: A Systematic Review

**DOI:** 10.3390/healthcare11091225

**Published:** 2023-04-25

**Authors:** Anna Nguyen, Valerie Eschiti, Thanh C. Bui, Zsolt Nagykaldi, Kathleen Dwyer

**Affiliations:** 1Fran and Earl Ziegler College of Nursing, University of Oklahoma Health Sciences Center, Oklahoma City, OK 73117, USA; 2Department of Family and Preventive Medicine, University of Oklahoma Health Sciences Center, Oklahoma City, OK 73117, USA; 3TSET Health Promotion Research Center, Stephenson Cancer Center, University of Oklahoma Health Sciences Center, Oklahoma City, OK 73104, USA

**Keywords:** Vietnamese, mobile health (mHealth) interventions, health behaviors, health service research, systematic review

## Abstract

The purpose of this review is to summarize the feasibility, acceptability, and efficacy of interventions that utilize mobile health (mHealth) technology to promote health behavior changes or improve healthcare services among the Vietnamese population. Ovid MEDLINE, CINAHL, EMBASE, Scopus, and Web of Science were used to identify studies published from 2011–2022. Studies utilizing mHealth to promote behavior change and/or improve healthcare services among Vietnamese were included. Studies that included Vietnamese people among other Asians but did not analyze the Vietnamese group separately were excluded. Three independent researchers extracted data using Covidence following PRISMA guidelines. Measures of feasibility, acceptability, and efficacy were synthesized. The ROBINS-I and RoB2 tools were used to evaluate methodological quality. Fourteen articles met inclusion criteria and included 5660 participants. Participants rated high satisfaction, usefulness, and efficacy of mHealth interventions. Short message service was most frequently used to provide health education, support smoking cessation, monitor chronic diseases, provide follow-up, and manage vaccination. Measures of feasibility, acceptability, and efficacy varied across studies; overall findings indicated that mHealth is promising for promoting lifestyle behavior change and improving healthcare services. Cost effectiveness and long-term outcomes of mHealth interventions among the Vietnamese population are unknown and merit further research. Recommendations to integrate mHealth interventions are provided to promote the health of Vietnamese people.

## 1. Introduction

As technology has advanced, mobile health (mHealth) has been increasingly used in the context of health promotion and disease management. Mobile phones, which are widely utilized for work, recreation, and communication, are also becoming widespread tools for health education, health promotion, and chronic disease management in multiple populations. Mobile phone use is now omnipresent, with 93% of the world population having access to a mobile-broadband network [1]. Easy access to devices coupled with wide network coverage allows mHealth to further expand its utilization.

The World Health Organization (WHO) classifies mHealth as the use of mobile phone applications, personal digital assistants, patient monitoring equipment, and other mobile wireless technologies in healthcare and also recognizes that such technologies can play significant roles in reducing the morbidity and premature mortality from noncommunicable diseases, as well as increasing access to health services [2]. The features and functions of mobile devices allow for quick access to information and enhanced communication, while meeting a variety of user needs [3]. Since most people own a mobile phone, mobile technology provides opportunities for healthcare providers to connect with patients. 

While the use of mobile technologies to improve health outcomes and health services is expanding rapidly among the Vietnamese population, the full extent of their utilization remains unclear due to the broad array of health-related services. Available mHealth interventions and their ongoing quality improvement necessitate a systematic review of mHealth usability and health outcomes in this specific population. Furthermore, demonstrating clear utilization of mHealth interventions among the Vietnamese population will guide future mHealth strategies to be highly engaged in changing health behaviors and improving health outcomes. At the time of writing, there has been no systematic review conducted to explore the acceptability, feasibility, and efficacy of mHealth interventions in promoting healthy behaviors or improve healthcare services among the Vietnamese population. In this systematic review, we aim to summarize the feasibility, acceptability, and potential efficacy of interventions that utilize mHealth technology to promote health behaviors or to improve healthcare services in Vietnamese individuals. Findings from this review allow for a broader understanding of mHealth research and utilization in this particular population and are useful for directing future development of mHealth interventions to increase their effectiveness. 

## 2. Materials and Methods

### 2.1. Data Sources and Study Selection

This review was conducted in accordance with the Preferred Reporting Items for Systematic Reviews and Meta-Analysis (PRISMA) guidelines [4]. Because of the study heterogeneity and the limited number of comparable studies that reported effect estimates with a measure of variance among the Vietnamese population, a meta-analysis could not be performed. The search was conducted in August 2022 using MEDLINE (via Ovid), EMBASE (via Ovid), CINAHL, Scopus, and Web of Science databases in collaboration with a medical librarian. Search results were imported into the Covidence platform, a web-based software platform for systematic review management, and duplicates were removed. Studies with a publication date from 1 January 2011–1 July 2022 were included, with the language restricted to English. Articles dating from 2011 were included because during this year, the World Health Organization (WHO) identified a compendium of emerging health technologies and indicated their potential for being low-resource solutions for unmet medical needs [5]. Both free-text search and controlled vocabulary were used with multiple key terms, including Vietnam* or Viet nam*, telehealth* or tele-health*, mHealth* or m-health*, ehealth* or e-health*, telemed* or tele-med*, teleconsult* or tele-consult*, mobile health*, mobile* or portable* or software*, app*, cell phone* or mobile phone*, and smart phone* or smartphone*. 

Studies were eligible for review if they examined the feasibility, acceptability, and/or efficacy of interventions involving mobile technologies that enabled remote monitoring or delivering information through functions and applications such as telephone calls, text messages, automatic voice calls, and smartphone applications. Articles were initially screened by reading the title and abstract to determine eligibility. Then, relevant articles were included for full-text review. Authorship and journal names were not blinded. All analyses in this review were from published research; therefore, no ethical approval or participant consent was required.

### 2.2. Inclusion and Exclusion Criteria

Inclusion criteria were original studies that used mHealth as a component of the intervention delivery for promoting behavior change and/or improving healthcare services in Vietnamese populations. Interventions involving mHealth alone and multicomponent interventions with mHealth were included. All types of mobile technology and its utilization for health care interventions among Vietnamese populations were included. There were no limits on the age of study participants or study settings. Embracing variation in the mHealth devices and various types of mHealth interventions, study settings, and participants across the lifespan was necessary for an inclusive view of mHealth use. To maximize the ability to find all relevant publications, feasibility and pilot studies were included if they met the inclusion criteria. 

Exclusion criteria included intervention studies that sampled participants of Asian descent including Vietnamese individuals but did not evaluate or analyze this group separately; studies that evaluated Vietnamese participants but did not clearly describe methods and results; and studies that used mobile devices for data collection only. Studies that utilized mobile technology but were not related to health, even if they used apps, short message service (SMS), or calls, were excluded. Study protocols, poster presentations, case studies, review articles, commentaries, and editorials were also excluded.

### 2.3. Data Extraction and Data Synthesis

To minimize selection bias and information bias, three researchers performed title and abstract screening independently. Two researchers performed data extraction of all included articles, while results were independently reviewed and confirmed by the senior researcher (K.D.). Discrepancies were discussed and resolved with consensus agreement. Using Covidence, the following data were extracted: author, year, country, study design, type of device and mHealth intervention, study aim, duration, and findings on feasibility, acceptability, and efficacy (Table 1).

Two different assessment tools were utilized to assess risk of bias: Risk of Bias in Non-Randomized Studies of Interventions (ROBINS-I) and Revised Risk of Bias (RoB 2) [6,7]. The ROBINS-I uses the Cochrane-approved risk of bias approach and focuses on study internal validity, which assesses bias in seven domains, and the RoB 2 is a revised Cochrane risk of bias tool for randomized controlled trials (RCT), which assesses bias in five domains. Each domain of non-RCT studies (*n* = 9) was rated according to its risk as “low”, “moderate”, “serious”, “critical”, or “no information”. Each domain of RCT studies (*n* = 3) was rated according to its risk as “low”, “some concerns”, or “high”. Then, an overall risk-of-bias rating was given for each study based on the overall risk judgment criteria of each assessment tool. Similar to the process of data extraction, two researchers assessed the risks and consulted the senior researcher to resolve any discrepancies. Two articles were excluded from the risk of bias assessment, as the assessment domains are not pertinent for qualitative or mixed methods studies [8,9]. 

## 3. Results

### 3.1. Study Characteristics and Quality

A total of 1087 articles were retrieved and assessed for eligibility. After removal of duplicates, 555 articles were screened. Based on screening of titles and abstracts, 508 articles were excluded because the studies did not aim to promote health behavior changes or to improve health outcomes or healthcare services. The remaining 47 articles were reviewed further. Of these, an additional 33 articles did not meet the eligibility criteria. Fourteen articles met the inclusion criteria. All 14 studies were conducted in Vietnam. Of note, ten United States-based studies included Vietnamese participants in their sample of Asian Americans; however, full-text review showed that the data on Vietnamese participants were not analyzed separately, and so these ten studies were excluded from this review (Figure 1). 

The fourteen eligible studies involved 5660 participants, excluding children under one year old [10]. Participant ages ranged from 21–61 years; the mean age was over 30 years in 10 studies that reported age. Sample size also varied widely, from 8 to 1433 participants. Only three studies were RCTs (Table 1); the remaining studies were prospective cohort feasibility pilot studies (*n* = 5), cross-sectional studies (*n* = 3), a quasi-experimental study (*n* = 1), a mixed methods study (*n* = 1), and a qualitative study (*n* = 1). The cross-sectional studies were surveys of mHealth-based service recipients/users to assess usability and acceptability. Of these, a control or comparison group was present in four studies; only two studies included theoretical frameworks to guide intervention development, including the transtheoretical model, social cognitive theory, and cognitive behavioral theory [11,12]. 

Participant recruitment and delivery of interventions took place mainly in hospitals, clinics, community health centers, or community-based organizations. The targeted participants and mHealth interventions varied significantly depending on the studies’ aims/purposes. Six studies intended to improve health behavior changes, including treatment adherence [13,14], mental health care [15], and smoking cessation [11,12,16]. Eight studies intended to improve healthcare services, including medical records management [17], health monitoring and support [9,18,19], vaccination management [10], patient-provider communication [8], and follow-up care [20,21].

Using the ROBINS-I and RoB 2 risk of bias assessment tools, all studies were judged as having either moderate risk, serious risk, or some concerns. Nine non-RCTs were rated with “moderate” or “serious” risk of bias, two RCTs were rated with “some concerns”, and one RCT had “low” risk of bias. The moderate or serious risks in the non-RCTs were of selection bias among study participants who were established patients of healthcare facilities; three studies had only male participants [11,16,21]. The bias in selection excluded eligible participants who did not have established healthcare services within these organizations, as well as female participants. 

Two RCTs had deviation in intervention bias and bias arising from the randomization process due to the lack of effort in dealing with issues relating to baseline differences (i.e., age and gender representation), which could affect the internal validity. This level of bias risk among included studies carried threats to the generalizability of the findings. Additionally, the lack of control groups lessens the confidence of the results. Therefore, strong conclusions on the intervention feasibility, acceptability, and efficacy of mHealth could not be drawn. A summary of the risk of bias assessment for the included studies is provided in Table 2 and Table 3.

**Table 1 healthcare-11-01225-t001:** Characteristics and findings of studies that met inclusion criteria.

Author (Year)	Study Design (Participants)	Device—mHealth Intervention	Study Outcomes	Findings		
				Feasibility	Acceptability	Efficacy
McBride, B. et al. (2018) [8]	Qualitative—document review; observations; focus group discussions; in-depth interviews (*n* = 60)	Mobile phone— SMS	Improve access to maternal, newborn, and child health services and health equity utilizing mHealth intervention.	Not measured.	Participants reported satisfaction with SMS and willingness to pay a fee for service.	Increased knowledge, effective behavior change, communication, husband involvement, and strengthened relationships between participants and community health workers.
Vu, L. T. H. et al. (2016) [9]	Prospective cohort (mixed methods) (*n* = 411 for baseline survey; *n* = 482 for post-intervention survey)	Telephone— Hotline, SMS, and map of health services providers	Impact of the 12-month mHealth intervention on changes in knowledge and practices related to sexual and reproductive health among female migrants.	Ability to recruit and retain 411 participants with various backgrounds and demographics.	Participants rated SMS service as useful (64.9%, *n* = 288) and very useful (20.3%, *n* = 90).	Women’s knowledge of sexual and reproductive health increased by 70.3%, and sexual and reproductive health practices were improved by 85.5%.
Nguyen, N. T. et al. (2017) [10]	Prospective cohort (pre- and post- uncontrolled study) (*n* = 11,449)	Mobile phone— SMS	Impact of SMS reminders to improve the immunization program by increasing vaccination rate.	SMS reminders have been shown to improve immunization coverage and timeliness of vaccination.	93.3% (111/120) of interviewees were willing to pay for SMS reminders for immunization schedule.	Immunization rate of children under one year old increased significantly from 75.4% in 2013 to 81.7% in 2014 and to 99.2% in 2015.
Ngo, C. Q. et al. (2019) [11]	Randomized, cross-sectional study (*n* = 469)	Telephone—SMS, phone calls/counseling	Impact of national telephone counselling for smoking cessation (self-report quit rate at baseline, 7-day, and 6-month abstinence) and factors associated with the Quitline use.	Response rate of 28.4% (469/1648) after excluding callers who did not set counseling appointments.	88.5% of participants were satisfied with program. Satisfaction and engagement were factors associated with increased Quitline use.	Most participants felt more confident about quitting (74.3%) and took early action via their first quit attempt (81.7%); 18.3% reported more than 7-day abstinence period.
Jiang, N. et al. (2021) [12]	RCT (2 arms) (*n* = 100)	Mobile phone— SMS	Feasibility, acceptability, and preliminary efficacy of a fully automated bidirectional SMS smoking cessation 6-week intervention.	Recruitment rate of 99% (100/101) enrolled in program and completed 12-week follow-up survey. In-depth interviews were also conducted to evaluate feasibility.	98% of participants in the intervention arm reported being satisfied with the program versus 82% in the control arm.	Biochemically verified abstinence was higher in the intervention arm at 6 weeks (20% vs. 2%), but the effect was not significant at 12 weeks (12% vs. 6%).
Nguyen, T.A. et al. (2017) [13]	Prospective cohort (uncontrolled feasibility study) (*n* = 40)	Smartphone—SMS, participants record themselves taking treatment and upload video to online server	Feasibility of using asynchronous Video Directly Observed Therapy (VDOT) to support treatment adherence among patients with pulmonary tuberculosis for 12 months.	51% (40/78) participated and rated the VDOT as feasible and interface highly, despite facing some initial technical difficulties.	87.5% (*n* = 35) found that VDOT was easy to use and stated they would recommend this service to others.	71.1% (*n* = 27) of participants took all required doses. A median of 88.4% of doses were correctly recorded and uploaded. 85% (*n* = 34) of participants missed <4 video uploads during the follow-up period.
Tran, B. X. & Houston, S. (2012) [14]	Cross-sectional survey (*n* = 1016)	Mobile phone— SMS, direct phone calls, and automatic voice calls	Feasibility of using mobile phone to support antiretroviral treatment adherence for patients with HIV/AIDS.	Expressed preferences for SMS (41.8%), direct calls (35.4%), direct counseling (43.1%), automated pill taking reminders (29.1%), regular information messages (21.3%), and clinic visits booking (16.5%).	63.5% of participants were willing to use services and willing to pay a fee for SMS adherence support service.	Majority of participants (78.6%) considered using mobile phone could be an effective adherence support.
Imamura, K. et al. (2021) [15]	Randomized controlled trial (RCT) (3 arms) (*n* = 951)	Smartphone—Smartphone application	Effect of a 10-week smartphone-based internet cognitive behavioral therapy stress management program to improve depression and anxiety among nurses.	Recruitment rate of 75.8% (962/1269) participated in baseline survey; 90% completed 7-month follow up for all 3 groups.	Completion rates (84%), satisfaction (>82%), and usefulness (>80%) in both intervention groups.	Depression and anxiety average scores decreased at 3 months from baseline but increased again at 7 months from baseline in both intervention groups.
Huang, W.-C. et al. (2022) [16]	Prospective cohort (uncontrolled feasibility single-arm study) (*n* = 221)	Mobile phone— Telephone calls, short message service (SMS)	Feasibility of a 12-month smoking cessation intervention that integrates follow-up counseling phone calls and scheduled text messages with brief advice from physicians.	Of 431 who were eligible, 221 (51.3%) agreed to participate in program.	141 (63.8%) participated in all 4 phone calls; 117 (52.9%) participated in all 8 phone calls in first 30 days.	90 (40.7%) self-reported abstinence from smoking in previous 30 days. Overall, 5.9% of all participants achieved verified smoking cessation for more than 30 days 12 months after enrollment.
Tran, B. X. et al. (2018) [17]	Cross-sectional study (*n* = 429)	Smartphone— application for vaccination management	Efficacy, adoption, and feasibility of implementing an mHealth application to educate and deliver information about vaccination and immunization.	Ability to recruit 429 participants with different levels of socio-demographic background.	Participants reported willingness to use (90.1%) and willingness to pay for the app 79.1%).	69.6% of participants believed that the app was necessary. Those who thought the app was unnecessary also felt there was sufficient vaccination information available online.
Khanh, T. Q. et al. (2020) [18]	Prospective cohort (uncontrolled pilot single-arm study) (*n* = 279)	Smartphone— Mobile app, SMS	Improve glycemic control and user satisfaction of incorporating a 12-week digital diabetes care system that monitor patient data and adjust therapy through digital contact.	Recruitment rate of 93% (279/300) participation. At week 12 and during the 20-day follow-up period, 81% remained engaged with the system and maintained glucose monitoring.	Both patients and healthcare professionals completed questionnaires at the final visit and reported overall satisfaction with system.	79% of participants had decreased average glucose levels, 36.9% of participants had decreased fasting glucose in first 2 weeks and last 2 weeks, and 45% of participants had HbA1c decreased from baseline at 12-week follow up.
Nguyet, T.T. et al. (2021) [19]	Quasi-experimental with a nonequivalent control group design (*n* = 52)	Smartphone, tablet, personal computers—SMS, viewings of educational content	Effect of a 4-week newborn care education program on breastfeeding rate and maternal role confidence of first-time mothers.	69% (70/101) agreed to participate with an attrition rate of 72% in the control group and 78% in the experimental group.	Not measured	At 4 weeks postpartum, the experimental group showed a significantly higher level of breastfeeding rate (*p* < 0.05) and mean maternal role confidence (*p* < 0.05) than the control group.
Ngoc, N. T. N. et al. (2014) [20]	RCT (*n* = 1433)	Telephone— Phone follow-up calls	Feasibility, acceptability, and efficacy of a service delivery protocol that replaces the routine clinic visit after medical abortion.	Phone follow-up offers a feasible approach to review pregnancy test result and checklist responses with the participants.	Most participants (88.3% [606/686]) indicated preference to have phone call follow-up from a healthcare provider.	Phone call follow-ups enable 85% of women to avoid a routine clinic visit without any decrease in safety.
Shapiro, L. M. et al. (2021) [21]	Prospective cluster (uncontrolled feasibility pilot study) (*n* = 8)	Mobile phone— SMS reminders and follow-up data collection	Feasibility of a 12-week SMS follow-up to obtain patient-reported outcome measures after hand surgery.	100% (8/8) were eligible and agreed to participate with 100% attrition.	Majority (>75%) of patients completed follow-up questionnaires at all data collection points.	SMS may serve as an effective method for follow-up to ensure safety and quality healthcare in low-resource settings.

### 3.2. Feasibility of mHealth Interventions

The majority (9/14; 64%) of studies included were designed to evaluate feasibility with outcome measures on recruitment, engagement, and retention. Six studies reported recruitment rates ranging from 51% to 99% [12,13,14,15,16,18]. These investigators attributed their high rates of recruitment and engagement to the modification and cultural adaptation of the mHealth interventions using participants’ suggestions as well as administrative support of project activities from top leaders.

In a study regarding the use of mHealth to provide video directly observed therapy in support of treatment adherence for people with tuberculosis in Vietnam, Nguyen et al. [13] reported the lowest recruitment rate of 51% and noted that nearly half of eligible patients did not agree to participate despite the free smartphone and technical support. Reason(s) for non-participation were not reported; however, the investigators suggested that mHealth usage for treatment adherence in this patient population is still feasible. Two other studies showed that mHealth interventions were feasible for ethnic minority patients in Thai Nguyen Province of Vietnam and for those with unequal access to services and health information [8,9].

Two studies reported engagement and retention rates ranging from 69% to 100%, but did not report recruitment rates [9,19]. These investigators attributed their high engagement and retention rates to the intensive reminders sent by their support staff using SMS, online chat group, and hotline telephone services. Although there was a gradual decline in patient engagement over the 12-week intervention period, glucose and HbA1c levels were significantly lower after the intervention [18]. The investigators attributed this finding to potential learning, information retention, lifestyle adjustment, and medication regimen adherence by the subjects.

Some studies sought to improve healthcare services. For example, Ngoc et al. [20] measured feasibility based on engagement with participants through scheduling follow-up calls, ability to obtain test results, and checklist responses via phone calls. Similarly, Shapiro et al. [21] reported 100% participation and retention using SMS for follow-up after hand surgery. In a prospective cohort study, Nguyen et al. [10] registered parents of 11,449 children born in Ben Tre province between 2013–2015 to receive SMS reminders about their child’s immunization schedule. These investigators reported that the on-time vaccination rates of BCG, measles, and Quinvaxem vaccines increased over time and that these rates maintained even after the project ended.

In general, the analyzed studies found that utilizing SMS to send reminders, health education information, follow ups, and/or healthcare support to participants is feasible. Additionally, utilizing SMS to engage with patients and send targeted motivational messages will likely be successful to both educate recipients and support chronic disease management and treatment adherence. Improving health literacy is critical for patients to make decisions related to self-management of chronic illnesses [22].

### 3.3. Acceptability of mHealth Interventions

While not all studies assessed acceptability, those that did used outcomes on perceived usefulness, satisfaction, participation, and willingness to pay for mHealth services. Four (29%) studies used self-report questionnaires to assess satisfaction, usefulness, and participation. Imamura et al. [15] asked participants to rate program satisfaction at a 3-month follow-up using a Likert scale that ranged from “very satisfied” to “very dissatisfied” and to rate usefulness of intervention at 3- and 7-month follow-ups using a scale ranging from “very useful” to “very useless”. “More than 80% of participants provided ratings, and most respondents rated the program with “somewhat satisfied” and the intervention as “quite useful”. These investigators attributed their high level of acceptance to the culturally tailored program with relevant wording and illustrations based on input from the target population. Ngoc et al. [20] asked for participants’ preferences for phone call or onsite clinic visit as a method for follow-up appointments; 88.3% preferred phone calls. Huang et al. [16] reported acceptability based on the ability to reach participants during follow-up phone calls; 141 (63.8%) participants answered four calls, and 117 (52.9%) answered all eight calls within the first 30 days of the Quitline program. Shapiro et al. [21] used SMS as reminders for patients to complete the Patient-Reported Outcome Measures (PROMs) instrument via a link following hand surgery and reported that >75% of patients completed the follow-up 11-item questionnaire at 1 day, 1 week, 2 weeks, 4 weeks, and 12 weeks post-surgery.

Nine (64%) studies obtained both qualitative and quantitative data to assess acceptability and user satisfaction. Jiang et al. [12] and Vu et al. [9] conducted in-depth interviews and explored perceptions about the overall program and specific characteristics, such as wording, messages, and timing of SMS delivery. They found that most participants liked the SMS and described it as helpful and useful. Khanh et al. [18] reported patient and provider satisfaction with the digital diabetes care system, and Nguyen et al. [13] reported patient satisfaction with the video directly observed therapy to support pulmonary tuberculosis treatment adherence post-intervention. Others reported high satisfaction with frequency and timing of SMS messages (two to four per day at participants’ preferred time) and that participants were willing to pay a fee for this service [8,10,14,17].

Overall, participant satisfaction was high, and participants perceived SMS interventions to be an acceptable and useful method. For studies reporting intervention frequency, those using frequent interactions with participants (i.e., two to four SMS messages daily) reported higher acceptability than did those with less frequent engagement (weekly SMS messages). Many studies reported the willingness of participants to pay a fee for SMS; however, the significant differences in the cost (range from $0.50 USD per month to $9.00 USD per smartphone app) and type of service make it impossible to draw definitive conclusions regarding acceptability based on this outcome. Overall, the concepts used to measure acceptability differed widely among the included studies, suggesting a need for standardized definitions, assessments, and methods for reporting acceptability of mHealth interventions.

### 3.4. Efficacy of mHealth Interventions

Seven (50%) studies measured efficacy based on participant self-report questionnaires and/or interviews. For instance, Huang et al. [16] used self-report data and found that 40.7% of participants abstained from smoking and 73.8% had at least one attempt to quit within the previous 30 days. Imamura et al. [15] assessed depression and anxiety symptoms at baseline, 3 months, and 7 months and found that symptoms decreased at 3 months but increased again at 7 months based on self-report data. Vu et al. [9] reported increased knowledge of sexual and reproductive health and improved healthy practices at 12 months post-intervention. Similarly, increased knowledge, effective communication and engagement among participants and their husbands, and strengthened relationships between participants and community health workers were measures of efficacy using qualitative interviews that yielded rich data to better understand how mHealth technology impacts behaviors [8]. Other studies indicated that SMS may serve as an effective method for medication adherence support [14] and for patient follow-up care to ensure safety and quality healthcare in low-resource settings [20,21].

Only four studies reported objective criteria as outcome measures of intervention efficacy, in addition to self-report data. Jiang et al. [12] used biochemical verification of smoking abstinence and found that efficacy was higher at 6 weeks in the intervention group, although efficacy was not significant at 12 weeks. Khanh et al. [18] reported the efficacy of a digital diabetes care system, which helped decrease the average glucose level by 11.5% and HbA1c level by 8.4% from baseline. Nguyen et al. [10] compared full immunization rate, immunization dropout rate, and timeliness of vaccination before and after ImmReg intervention and found that immunization rates for children under one year of age increased significantly from 75.4% before the intervention to 81.7% immediately after and to 99.2% one year after intervention (*p* < 0.01). Nguyet et al. [19] compared participant responses pre- and post-intervention and found that the intervention group showed significantly higher rates of breastfeeding and levels of maternal role confidence in first-time mothers (*p* < 0.05) than did the control group.

## 4. Discussion

In this systematic review, we identified, appraised, and synthesized 14 studies that evaluated the feasibility, acceptability, and efficacy of mHealth interventions for the Vietnamese population. Six studies aimed to improve health behavior changes, categorized as (a) treatment adherence, (b) mental health care, or (c) smoking cessation. Eight studies aimed to promote healthcare services, specifically (a) medical records management, (b) disease monitoring and support, (c) immunization reminders, (d) patient-provider communication, or (e) follow-up care. This is the first systematic review to compile 11 years of mHealth intervention studies that focused on improving health behaviors or healthcare services for Vietnamese people.

Devices utilized were smartphones, mobile phones, mobile tablets, personal computers, and medical devices connected to phones via cloud-based software (see Table 1). Devices were used for patient remote monitoring, assessment, or counseling for health behavior change through a wide range of functions and applications, such as text messaging, internet access, email, and videos. SMS was most frequently used (11/14; 79%) as the primary intervention, which is consistent with many studies that evaluated the utilization of mHealth to support lifestyle and health behavior changes in other populations [23,24,25,26]. SMS was used to provide health education (*n* = 3; 21%) disease treatment, support smoking cessation (*n* = 3; 21%), monitor and support chronic diseases (*n* = 3; 21%), follow up (*n* = 1; 7%), and manage vaccination (*n* = 1; 7%). These investigators referred to the low cost and easy operation of SMS, which requires a low level of technical skills to receive and send messages. Other studies incorporated multiple components of mHealth, such as combining SMS with telephone calls [14,16] or SMS with phone counseling and printed materials [9].

Despite the type of device or applications utilized, it was noted that offering frequent interaction with users was necessary to engage participants and sustain the newly adopted health behaviors. Frequency of intervention delivery varied; however, our review shows that sending two to four SMS messages per day could have potentially positive impacts on lifestyle behavior change or on the delivery of healthcare services among Vietnamese individuals. Intervention duration ranged from 1 week to 12 months, with an average duration of approximately 17 weeks. Five studies reported only immediate post-intervention outcomes, while four others reported additional follow-up periods of 20 days, 30 days, 2 months, and 12 months, respectively [10,11,16,18].

Most of these studies were based on subjective data such as self-reported questionnaires, underlying the need for more objective measurement in future research. Furthermore, twelve studies (86%) did not integrate a theoretical framework. Lack of a theory or framework makes it difficult for readers to clearly discern the investigators’ assumptions underlying the study methodology [27]. Having a theoretical framework helps organize the concepts and constructs, and its precepts help guide the study [27]. Two studies that focused on smoking cessation incorporated theoretical frameworks and framed the interpretation of their findings accordingly.

To optimize delivery and efficacy of mHealth interventions, behavioral frameworks may be applied in addition to psychological models. For instance, applying the Technology Acceptance Model may contribute to increasing technology acceptance and adoption. Venkatesh and Davis [28] posited that the intention to use new technology is determined by perceived ease of use and perceived usefulness. Furthermore, attitudes toward the technology also influence the decision to use [29]. Therefore, understanding and incorporating the determinants of intention to use and attitudes toward the new technology adoption in the intervention design are more likely to increase acceptance and usage.

Another consideration in the design of effective interventions may include the application of the Supportive Accountability Model to support mHealth intervention adherence with human support [30] and applying the models of behavioral change to trigger and motivate health behavior changes [31,32]. Furthermore, applying a multi-channel approach including detached co-involvement to strengthen the relationship between healthcare professionals and patients receiving outpatient mental health care through frequent digital interactions may increase patient autonomy and strengthen patient-provider relationships.

Based on the assemblage theory [33], the integration of mHealth technology can extend care assemblages temporally and spatially. Schneider-Kamp and Fersch [34] revealed evidence of improved care processes, care outcomes, and care relations between healthcare professionals and mentally vulnerable patients while potentially increasing their autonomy. Moreover, technological acceptance by end users may be enhanced with relational trust between the two parties. Finally, applying domestication theory, particularly incorporation and conversion dimensions of technology appropriation, will facilitate understanding of how mHealth interventions could become a part of users’ everyday lives [35].

With a variety of mHealth intervention purposes and target groups, measurement of participant satisfaction varied and measurement of willingness to pay, with the amount either not specified or showing large variations between studies, may influence the ratings. These outcome measures were mixed on intervention acceptability, suggesting a need for more standardized methods of measuring and reporting these constructs. Furthermore, there is a growing interest in the utilization of mHealth technology to promote health behavior changes and improve healthcare services, cost effectiveness, and long-term outcomes of mHealth interventions among the Vietnamese population worldwide. Therefore, these factors will need to be tested with the use of theoretical models to better understand the mechanisms that affect adoption and effectiveness of mHealth technologies.

Barriers to and facilitators for implementing mHealth interventions were identified by the investigators of studies included in this review. Among the barriers, Khanh et al. [18] and Shapiro et al. [21] reported that approximately one quarter of participants felt that the lack of smartphones, inadequate internet connectivity, and cost for data use may hinder mHealth initiatives. Another barrier was the requirement for participants to call and sign up for the study, possibly explaining the lower-than-expected recruitment rate for these mHealth interventions [16]. Another drawback identified by Imamura et al. [15] was the low intervention effects among participants with depressive symptoms—fully automated and self-guided programs may make it difficult for these participants to engage in the program without personal interactions for support and advice. Hou et al. [36] found that mHealth interventions combined with professional healthcare provider management is essential to enhance clinical efficacy.

Several facilitating factors for the success of mHealth intervention implementation were also identified. The ability to recruit and retain participants was enhanced by support from organizational leaders and a collectivist culture valuing community needs over individual needs, which facilitated a low attrition rate. Participants having access to mobile phones and being familiar with SMS, a decreased clinic workload for healthcare providers, and individual privacy were also identified as facilitators. For instance, patients perceived privacy as an advantage of SMS over an onsite clinic visit after medical abortion, for HIV/AIDS treatment adherence support, and for sexual and reproductive health education [9,14,20]. These findings highlighted the need for cultural sensitivity of interventions that may be stigmatized in this population by reducing the need for in-person visits while also reducing communication delays.

Based on the barriers and facilitators described, several solutions could be considered for future research on mHealth interventions among Vietnamese participants. First, providing options for potential participants to sign up or to ask questions as they consider participating may facilitate recruitment. Second, combining mHealth interventions with healthcare professional interaction is recommended over exclusively mHealth approaches. Third, ensuring participant privacy and cultural sensitivity while providing reliable, authentic, and practical mHealth services is necessary. Finally, maintaining a strong commitment to community-level dissemination by engaging with local officials and legislators is also recommended [37].

Compared to other systematic reviews that examined the feasibility, acceptability, and efficacy of mHealth interventions among various populations and services, the outcomes suggested that mHealth intervention is feasible, acceptable, and effective for most participants [38,39,40,41,42,43]. These systematic reviews indicated that study participant satisfaction rates were often reported to illustrate feasibility and acceptability. Han and Lee [38] reported that 80% of reviewed studies (*n* = 16) demonstrated that mHealth applications positively impact health behavior changes, such as in physical activity, alcoholism, dietary changes, adherence to medication or therapy, and clinical outcomes. Buck et al. [40] also found that the majority of reviewed studies demonstrated that SMS and email interventions were acceptable and effective in improving adherence among study participants’ postoperative pain management.

Da Silva et al. [42] reviewed 19 articles and found that mHealth application was feasible and acceptable in monitoring patients with head and neck cancer and self-management of their conditions. While feasibility and acceptability were found in all six systematic reviews, mHealth intervention effectiveness was demonstrated in some studies. Abasi et al. [39] found that 62.5% of the reviewed studies (*n* = 10) demonstrated the use of mHealth to be effective in medication adherence and self-management among patients post transplantation. Wickershan et al. [41] reported that the use of mHealth showed evidence of feasibility and acceptability for delivery of interventions via mobile application for people with post-traumatic stress disorder; however, there was inconsistent evidence on its effectiveness. Grist et al. [43] reviewed 24 articles and found that mHealth use among children and adolescents was feasible and acceptable; however, there were 3 studies that did not demonstrate that mHealth applications were effective in improving mental health outcomes.

If an mHealth intervention is to be developed for the Vietnamese population to promote health behavior change and improve healthcare services, the degree of feasibility, acceptability, and efficacy must be understood. This systematic review differs from other reviews in that it focuses solely on the Vietnamese population, and it confirms that the feasibility, acceptability, and efficacy of mHealth interventions may be generalized to the Vietnamese population.

While the methodological risk of bias among included studies was moderate to serious and with some concerns, this systematic review highlights sound evidence that mHealth interventions are feasible, acceptable, and efficacious in improving healthy behaviors and healthcare services among the Vietnamese population. The review’s results regarding participant recruitment, engagement, and retention demonstrated high feasibility. The features of mHealth technology were found to be useful, with high satisfaction ratings indicating a high level of acceptability. However, the efficacy of interventions decreased over time in studies that followed participants long term. This finding indicates that mixed methods research is needed to measure the long-term impact and to understand the rationale for declining efficacy in long-term mHealth interventions with larger sample sizes.

### Strengths and Limitations

This systematic review has several strengths and limitations. The strengths were (1) the collaboration with a university librarian to conduct a comprehensive and robust search strategy, (2) the use of the Covidence platform to systematically conduct the study selection and data extraction process and assess each study quality accordance to the PRISMA guidelines, and (3) the use of the ROBINS-I and RoB 2 instruments to assess the risk of study bias. One limitation is that all studies were conducted in Vietnam, and, therefore, findings are not generalizable to Vietnamese people living outside of Vietnam. Other limitations are the inclusion of only studies published in English and indexed in a computerized database. Relevant studies may have been missed if they were published in Vietnamese or other languages or if the journals were not indexed in a widely accepted electronic database. Finally, although the literature search was extensive, a meta-analysis cannot be performed due to the heterogeneity of methods and the small number of studies included in this review. Therefore, any interpretation from this review cannot be generalized to a larger population to confirm the feasibility, acceptability, and efficacy of mHealth interventions.

## 5. Conclusions

With the increasing popularity of mobile technologies, a shift in healthcare to incorporate mHealth interventions is necessary to improve patient outcomes and satisfaction. This systematic review provided a detailed summary of evidence for the feasibility, acceptability, and efficacy of mHealth interventions to improve a broad range of health behaviors and services among the Vietnamese population and demonstrated that mHealth can play a key role in translating technology into improved patient outcomes. These findings are encouraging for future research using mHealth interventions to promote the health of Vietnamese people. However, with the limitations regarding both quality and quantity of reviewed studies, there is a pressing need for high quality, theory-driven, and multicenter trials, including studies outside of Vietnam, to substantiate and generalize these findings.

## Figures and Tables

**Figure 1 healthcare-11-01225-f001:**
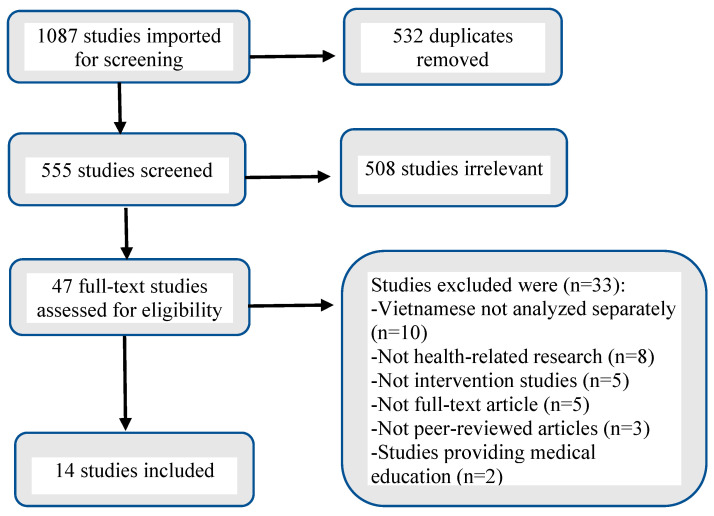
Flow chart of the literature search and study selection process.

**Table 2 healthcare-11-01225-t002:** Summary of study quality assessment—Risk of bias assessment for non-randomized controlled trials.

	Confounding Bias	Selection Bias	Intervention Classification Bias	Deviation in Intervention Bias	Missing Data Bias	Measurement Bias	Selection of Reported Results Bias	Overall Rating	
Nguyen, N.T. et al. (2017) [10]									Green = Low risk of bias Orange = Moderate risk Red = Serious risk Blue = Critical risk Gray = No information
Ngo, C.Q. et al. (2019) [11]									
Nguyen, T.A. et al. (2017) [13]									
Tran, B.X. & Houston, S. (2012) [14]									
Huang, W-C. et al. (2022) [16]									
Tran, B.X. et al. (2018) [17]									
Khanh, T.Q. et al. (2020) [18]									
Nguyet, T.T. et al. (2021) [19]									
Shapiro, L.M. et al. (2021) [21]									

**Table 3 healthcare-11-01225-t003:** Summary of study quality assessment—Risk of bias assessment for randomized controlled trials.

	Randomization Process Bias	Deviation in Intervention Bias	Missing Data Bias	Measurement Bias	Selection of Reported Results Bias	Overall Rating	
Jiang, N. et al. (2021) [12]							Green = Low risk of bias Orange = Some concerns Red = High risk
Imamura, K. et al. (2021) [15]							
Ngoc, N.T.N. et al. (2014) [20]							

## Data Availability

No new data were created. Data sharing is not applicable.

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
