# Peer review of "Mobile Health Interventions to Improve Health Behaviors and Healthcare Services among Vietnamese Individuals: A Systematic Review"

_healthcare, 2023, doi:10.3390/healthcare11091225_

Round 1
Reviewer 1 Report
Dear Authors,
first of all, I want to emphasize that the work of conducting a meta-analysis is a very time-consuming task and requires a lot of effort and adherence to fairly strict standards. in my opinion, you have chosen an interesting and relevant topic, and the conclusions drawn are based on evidence-based standards.
I would like to ask a few questions, write some recommendations that I hope will help you improve the manuscript
in the case of meta-analysis, it is worth noting when exactly you conducted a direct search of articles in the databases
In the methods, you should specify the search strings you have created (for example, for Medline)
Inclusion/exclusion criteria should be specified according to a specific system (for example, PICO that is gold standard for meta-analysis)
It is also worth justifying why articles dating from 2011 were included
Kind regards,
Reviewer 2 Report
The manuscript describes and discusses the results of a systematic review of the feasibility, acceptability, and efficacy of mHealth for health interventions among the Vietnamese population. The analysis seems rigorous, and the manuscript is generally well written.
There are two major concerns, though, which need to be addressed before the manuscript can be considered publishable.
1) The manuscript does not summarize and relate to systematic studies of the feasibility, acceptability, and efficacy of mHealth interventions among other populations. This is a blind spot that needs to be remedied, if we are to understand to what degree the findings are specific to the Vietnamese population and to what degree they can be generalized and transferred to other populations/contexts.
2) I commend the manuscript for pointing out and discussing the relative lack of theoretical foundations for the studies of feasibility, acceptability, and efficacy of mHealth for health interventions. The discussion of the manuscript would, however, be significantly strengthened by a short review of theories that have been applied or might be applied in the context of mHealth. This would provide immense value to further studies of this growing field. In addition to transtheoretical model, social cognitive theory, and cognitive behavioral theory, some further well-known theories are summarized in the following systematic literature review:
Chib, A., & Lin, S. H. (2018). Theoretical Advancements in mHealth: A Systematic Review of Mobile Apps. Journal of Health Communication, 23(10–11), 909–955. https://doi.org/10.1080/10810730.2018.1544676
This review explicates additional theories such as the technology acceptance model and the supportive accountability model in addition to models for behavioural change.
Beyond 2018, the authors might also strengthen the manuscript by including some emerging theories on the acceptability and efficacy of mHealth such as detached co-involvement based on assemblage theory and domestication theory from science and technology studies:
Schneider-Kamp, A., & Fersch, B. (2021). Detached co-involvement in interactional care: Transcending temporality and spatiality through mHealth in a social psychiatry out-patient setting. Social Science & Medicine, 285, 114297. https://doi.org/10.1016/j.socscimed.2021.114297
Presset, B., Kramer, J.-N., Kowatsch, T., & Ohl, F. (2021). The social meaning of steps: User reception of a mobile health intervention on physical activity. Critical Public Health, 31(5), 605–616. https://doi.org/10.1080/09581596.2020.1725445
A few minor detailed remarks:
Line 138: Please remove ”that” before ”did not meet”.
Line 177: Please insert “the” before “generalizability”.
Line 316: “The main mHealth device utilized was smartphones” sounds awkward. Maybe “devices utilized were”?
Round 2
Reviewer 1 Report
Dear Authors,
Thank you a lot for the clarification. Congratulation once more with great result!
Regards,
Author Response
Dear Reviewer:
Thank you for your kind wishes.
Reviewer 2 Report
The authors have made an attempt at addressing the concerns regarding related work and other theoretical perspectives and, thereby, moved the manuscript a bit closer to a publishable state.
Currently, however, the way that these concern were addressed have significant shortcomings that need to be addressed in another revision:
1) The references for related work on other populations and other contexts is relevant and strengthens the manuscript. However, the authors HAVE to unfold these references beyond the single summarizing sentence currently given in Lines 406-409. The reader is currently left with "[38-43]", i.e., with the insurmountable task of unpacking 6 references without any help by the authors. The authors also need to specify which outcomes the phrase "the outcomes" refers to.
2) The authors again fail to unpack the references. Saying that "applying the Technology Acceptance Model may contribute to increase technology acceptance" in Lines 353-356 is simply not sufficient. Also, some of the new sentences/sentence fragments do not make any sense - neither grammatical nor content-wise.
3) The authors have integrated a short reflection on how mHealth contributes to "to strengthen the relationship between healthcare professionals and patients through frequent interactions while increasing patient autonomy" in Lines 356-359. They do, however, wrongfully reference 33 (Ruiz et al. 2020) and 34 (Hoffman & Novak 2018) for this point and for the concept of "detached co-involvement". Neither of these articles discusses patient autonomy or detached co-involvement. The authors need to replace these two references with a relevant reference.
I wish the authors the best of luck with their revision!
Round 3
Reviewer 2 Report
The author's have successfully unfolded the references and addressed all open issues.